# The Combined Effects of Azoxystrobin and the Biosurfactant-Producing *Bacillus* sp. Kol B3 against the Phytopathogenic Fungus *Fusarium sambucinum* IM 6525

**DOI:** 10.3390/ijms25084175

**Published:** 2024-04-10

**Authors:** Aleksandra Walaszczyk, Anna Jasińska, Przemysław Bernat, Sylwia Różalska, Lidia Sas-Paszt, Anna Lisek, Katarzyna Paraszkiewicz

**Affiliations:** 1Department of Industrial Microbiology and Biotechnology, Faculty of Biology and Environmental Protection, Doctoral School of Exact and Natural Sciences, University of Lodz, 90-136 Lodz, Poland; aleksandra.walaszczyk@edu.uni.lodz.pl; 2Department of Industrial Microbiology and Biotechnology, Faculty of Biology and Environmental Protection, University of Lodz, 90-136 Lodz, Poland; anna.jasinska@biol.uni.lodz.pl (A.J.); przemyslaw.bernat@biol.uni.lodz.pl (P.B.); sylwia.rozalska@biol.uni.lodz.pl (S.R.); 3Department of Microbiology and Rhizosphere, The National Institute of Horticultural Research, 96-100 Skierniewice, Poland; lidia.sas@inhort.pl (L.S.-P.); anna.lisek@inhort.pl (A.L.)

**Keywords:** *Bacillus*, *Fusarium*, azoxystrobin, biosurfactants, surfactin, iturin, biofungicides, biocontrol, antifungal activity

## Abstract

This study aimed to evaluate how the combined presence of the synthetic fungicide azoxystrobin (AZ) and the biosurfactant-producing *Bacillus* sp. Kol B3 influences the growth of the phytopathogenic fungus *Fusarium sambucinum* IM 6525. The results showed a noticeable increase in antifungal effectiveness when biotic and abiotic agents were combined. This effect manifested across diverse parameters, including fungal growth inhibition, changes in hyphae morphology, fungal membrane permeability and levels of intracellular reactive oxygen species (ROS). In response to the presence of *Fusarium* and AZ in the culture, the bacteria changed the proportions of biosurfactants (surfactin and iturin) produced. The presence of both AZ and/or *Fusarium* resulted in an increase in iturin biosynthesis. Only in 72 h old bacterial–fungal co-culture a 20% removal of AZ was noted. In the fungal cultures (with and without the addition of the bacteria), the presence of an AZ metabolite named azoxystrobin free acid was detected in the 48th and 72nd hours of the process. The possible involvement of increased iturin and ROS content in antifungal activity of *Bacillus* sp. and AZ when used together are also discussed. Biosurfactants were analyzed by liquid chromatography with tandem mass spectrometry (LC-MS/MS). Microscopy techniques and biochemical assays were also used.

## 1. Introduction

Phytopathogens, including fungi, are one of the world’s most significant agricultural threats [1,2]. Those microorganisms have the capacity to cause devastating diseases in plants, resulting in significant economic losses, weakened food security and damage to the environment [1,2]. Issues such as the growing population, climate change and globalization make the interactions between pathogens, pesticides and the environment increasingly complex [1,3,4,5]. The understanding of phytopathogen interactions is critical for finding effective ways to reduce their impact on crop yields.

The fungal genus *Fusarium* includes many well-known, widespread, and highly adaptable plant pathogens, posing a threat to global agriculture [2,3,6,7]. These phytopathogens are known to often become resistant to synthetic fungicides, which are used to control them [7]. Therefore, *Fusarium* fungi have emerged as one of the most important contributors to severe crop diseases, causing significant economic losses and compromising global food security [2,7].

The significant threat posed by phytopathogens demands the usage of fungicides to protect agricultural productivity. Synthetic fungicides have historically played an essential role in preventing fungal infections and reducing the financial losses caused by them. However, in recent years, the use of these chemical agents has raised concerns regarding their environmental impact and the emergence of resistant phytopathogen strains [7]. Since some of the synthetic fungicides have a long half-life, there is a danger of them persisting in the environment for extended periods of time and having a large ecological impact [8]. Moreover, they can influence and alter the activity of soil microorganisms other than phytopathogens, including plant growth-promoting microorganisms (PGPM) such as *Bacillus* bacteria [9,10].Gram-positive bacteria from the genus *Bacillus* are characterized by their rod-shaped morphology and the ability to form endospores, which enables them to adapt to various environments (e.g., soil and water), even in extreme conditions (such as high temperatures, pH fluctuations or exposure to toxic substances) [11,12]. *Bacillus* bacteria are widely known to produce a variety of secondary metabolites of valuable biotechnological and industrial applications [11,13]. These secondary metabolites include antibiotics, enzymes, cyclic lipopeptide (cLP) biosurfactants, polyketides, and volatile compounds [14]. Their ability to synthesize antifungal compounds in particular has made them frontrunners in developing effective, environmentally friendly biofungicides—alternatives to conventional synthetic fungicides [11,13,14,15].

Some *Bacillus* strains are capable of producing cLP biosurfactants belonging to surfactin, iturin, and fengycin families [16,17]. Surfactin shows mainly antiviral and antibacterial properties, while iturin and fengycin exhibit mainly antifungal activity [14]. Importantly, studies have shown that biosurfactants produced by different bacilli and other antimicrobial compounds can act synergistically and enhance each other’s antimicrobial properties [18,19,20,21]. The antifungal mode of action of biosurfactants involves several different mechanisms, including disruption of cell membranes, alteration of membrane permeability, induction of apoptosis, and inhibition of spore germination or antibiotic activity (inhibition of fungal growth by, for example, interfering with cell wall synthesis) [21,22].

Even though biofungicides offer an environmentally friendly alternative to synthetic fungicides, they have several limitations, such as variability in efficacy and reliance on favorable environmental conditions [15]. A possible way of improving biofungicides is harnessing the combined potential of bio- and synthetic-based approaches to the problem. The combined use of biopesticides and synthetic pesticides presents a practical approach to pest management worldwide, allowing for obtaining the advantages of each type of pesticide. There have been reports of farmers integrating both types of pesticide to address pest resistance, optimize efficacy, and meet their production objectives [23,24,25,26,27]. This mixed-use strategy may allow for enhanced pest control while minimizing environmental impact and preserving biodiversity.

Azoxystrobin is a fungicide belonging to the strobilurin class, which has a soil half-life of 180 days in the field and is degraded by 90% after 600 days in the field [28]. AZ operates in the fungal mitochondrion, where it prevents electron transport and energy production [29]. This fungicide is commonly applied to protect many different varieties of plants, e.g., grains, fruits, and vegetables, and shows a wide range of activities against fungal infections from all four taxonomic groups: *Oomycetes*, *Ascomycetes*, *Deuteromycetes*, and *Basidiomycetes* [29]. According to Wang and co-workers’ findings [30], an ongoing AZ application can impact the soil microbiome, which in turn may inhibit important soil nutrient cycling in sustaining productive croplands.

The aim of the study was to compare the impact of AZ and biosurfactant-producing *Bacillus* sp. Kol B3, as well as these two agents combined, on *Fusarium sambucinum* IM 6525 mycelium. We also assessed the effect of AZ and the fungus on *Bacillus* sp. Kol B3 biosurfactant production.

## 2. Results and Discussion

### 2.1. Antifungal Activity

The influence of the combined application of AZ and *Bacillus* sp. Kol B3 on *F. sambucinum* IM 6525 mycelial growth in liquid cultures was investigated (Figure 1). The obtained results revealed that both *Bacillus* sp. Kol B3 and AZ strongly inhibited the growth of *F. sambucinum* IM 6525. Interestingly, it was established that in the 72nd hour of cultivation in the presence of either AZ, the bacteria, or both of them used in combination, the fungal growth reached 28.6 ± 5.9, 29.7 ± 3.6 and 15.3 ± 2.8%, respectively. Thereby, AZ and bacilli used simultaneously, after 72 h of the process, resulted in significantly stronger (*p* < 0.05) antifungal activity than when they were used alone.

Additionally, *Bacillus* sp. Kol B3 bacteria growth in the presence of AZ was assessed. It was found that AZ does not have a significant effect on bacterial growth (Appendix A).

The enhanced antifungal effect of bacteria from the genus *Bacillus,* when applied in combination with some synthetic pesticides, has been reported before [31,32,33,34,35]. For example, Peng and co-workers [31] studied a possible synergistic effect of *B. subtilis* NJ-18 with flutolanil and difenoconazole fungicides. The obtained results showed that, when combined, the agents demonstrated superior control of wheat sharp eyespot caused by *Rhizoctonia cerealis.* In another study, Liu and co-workers [33] found that the combined application of the biological agent *B. subtilis* H158 and synthetic fungicide strobilurin demonstrated a stronger inhibitory effect of rice sheath blight compared to strobilurin alone. Similarly, Chen and co-workers [35] showed that lipopeptide extracts from *B. velezensis* SDTB038, combined with the fungicide phenamacril, exhibited strong synergistic control against *F. oxysporum* f. sp. *radicis-lycopersici*, resulting in an 84% reduction in fungal growth.

Several possible mechanisms for the stronger antifungal effect of synthetic and biological control agents have been proposed. When combined with synthetic fungicides, biocontrol agents might be more successful in competing with the fungi for nutrients and space, which can lead to effective control of fungal phytopathogens [36]. Liu and co-workers [37] pointed out several factors that may be involved in the mechanisms of synergistic antifungal effect between the fungicide tebuconazole and *B. subtilis* H158.They suggested that the increased antifungal activity might be a result of tebuconazole stimulating *B. subtilis* H158 growth, enhancing bacterial persistence, promoting biofilm formation, regulating host defense mechanisms, and suppressing the pathogen’s natural resistance. In another study conducted by Liu and co-workers [33], the authors hypothesized that the synergistic effect of strobilurins and *B. subtilis* H158 could be a result of one of two possible mechanisms: (1) the strobilurins may facilitate the formation of *B. subtilis* H158 biofilms, or (2) the fungicide may increase oxidative stress, which may damage fungal cells and result in the excretion of nutrients that promote the growth of *B. subtilis* H158.

### 2.2. Microscopic Analyses

Laser scanning microscopic inspection revealed a lot of long hyphae with numerous branches in the control *F. sambucinum* IM 6525 cultures (Figure 2(a1)). The addition of AZ resulted in the formation of singular distorted fungal cells (Figure 2(b1)). As shown in Figure 2(c1), the additional presence of *Bacillus* sp. Kol B3 cells in the fungal culture led to visible mycelium fragmentation, and the integrity of the tip of the hyphal cells was disrupted. The co-cultivation of both microorganisms with AZ (Figure 2(d1)) resulted in the formation of very short fungal filaments of a larger circumference than was found in the control mycelium. Compared to the control, no organelles were visible in the hyphae. The observations made during the microscopic analysis confirmed the growth intensity results.

Next, the effect of the fungicide and the bacteria on *F. sambucinum* IM 6525 viability was assessed using the FUN-1 kit and the fluorescence microscope (Figure 2(a2–d2); Table 1). The viability of the mycelium in the control samples was 93.14 ± 6.45%, while the addition of the fungicide resulted in a significant reduction in fungal viability to 5.03 ± 1.49%. In the samples incubated with the bacteria, a further reduction in fungus viability to 0.18 ± 0.08% was observed. Of particular interest were samples to which the bacteria and the fungicide were added simultaneously. Under these conditions, all the inspected mycelia were found to be dead.

### 2.3. Effects of AZ and the Bacteria on Fungal Intracellular ROS Generation

AZ inhibits mitochondrial respiration, reducing ATP synthesis, which leads to oxidative stress in the target fungus [38]. Therefore, the intracellular level of peroxynitrate anions/hydroxyl radical anions in *F. sambucinum* IM 6525 cells exposed to the *Bacillus* sp. Kol B3 bacteria and/or AZ was determined (Table 2). The H_2_DCFDA assay was used as a non-specific indicator.

The obtained results revealed that the exposure to AZ alone significantly (*p* ≤ 0.05) increased intracellular ROS generation in the hyphae (3.99 ± 1.38% compared to 0.02 ± 0.03% in control culture), indicating the induction of oxidative stress within the fungal cells. The exposure of *F. sambucinum* IM 6525 to the bacteria resulted in an increase in ROS generation (0.77 ± 0.14%) which was moderate in comparison to AZ. Interestingly, the combination of the bacteria and AZ resulted in a very low (similar to the control) level of intracellular ROS generation in the fungus.

These results are consistent with the microscopic observations and the analysis of fungal cell viability and confirm the thesis about different mechanisms of action of AZ and *Bacillus* sp. Kol B3 on the phytopathogenic fungus *F. sambucinum* IM 6525. The surprisingly low levels of intracellular ROS observed in bacterial–fungal co-culture amended with AZ might result from strong damage to the mycelium, lysis of cell walls, and degeneration of the energy machinery of the cells [29,39]. A similar effect was observed by Guirao-Abad and co-workers [40] when they investigated the influence of two antifungal factors (amphotericin B 43 and micafungin) on *Candida albicans* cells. Both tested factors strongly limited yeast growth; however, increased ROS production was observed only in the case of the addition of amphotericin B 43, while the addition of micafungin (despite a clear fungicidal effect) did not increase the production of intracellular ROS. This was probably due to the high toxicity of the tested agent on the *C. albicans* cells. 

### 2.4. Permeabilization of Fungal Membranes

In order to unravel the mechanisms of combined antifungal activity of AZ and *Bacillus* sp. Kol B3, fungal membrane permeability was assessed (Figure 3).

Without the addition of either AZ or the bacteria, fungal membrane permeability was determined to be 266.8, 171.1, and 455.6 fluorescence intensity/biomass in the 24th, 48th and 72nd hours of cultivation, respectively. The presence of each additive (the bacteria and AZ used alone or in combination) to the fungal cultivation caused a significant increase in *F. sambucinum* IM 6525 membrane permeability. It was established that AZ addition to the fungal cultivations caused the fungal membranes to be 2.4 times more permeable at every investigated stage of cultivation. The *Bacillus* sp. Kol B3 bacteria proved to be even more effective than AZ (*p* ≤ 0.05). As soon as in the 24th hour of cultivation, the fungal membrane permeability reached 1591.1 fluorescence intensity/biomass (6.0 times more than that of control cultivation). The highest fungal membrane permeability was noted for the 72 h fungal cultivation containing both AZ and bacteria (1957.0 fluorescence intensity/biomass).

Previous studies had reported changes in the fungal membrane permeability caused by different pesticides [41,42]. AZ does not, however, directly affect the permeability of fungal membranes. Its primary target is the mitochondrial respiratory chain and disruption of energy production [29]. The results presented in this study indicate that AZ also increases the permeability of fungal membranes, which might be caused by increased ROS levels, damaging cell membrane components.

Contrary to AZ, the mode of action of *Bacillus* lipopeptides such as surfactin and iturin involves the disruption of fungal membranes by forming pores or affecting the membrane fluidity [21,22,42,43]. These changes weaken the membrane’s structural integrity, which limits the fungal pathogen’s ability to survive and reproduce.

Considering that AZ operates by preventing energy production in mitochondria [29] while biosurfactants’ main antifungal mechanism of action is the disruption of cell membranes [21,22], it seems reasonable to speculate that the enhanced antifungal effect stems from both of these agents using different modes of action.

### 2.5. Biosurfactant Production

It was found that surfactin levels decreased with time in every cultivation variant (Figure 4a). The obtained data demonstrated that the addition of AZ to the bacterial culture had no significant effect on surfactin production. A significant (*p* ≤ 0.05) decrease (at least 50%) in surfactin production was noted in the bacterial–fungal co-cultures. Interestingly, a mitigative effect of AZ on the decreased surfactin levels was observed. For example, compared to the production of surfactin in a 72 h bacterial cultivation (the control culture), the level of biosurfactant in the presence of mycelium decreased by approximately 60%; however, in the cultivation containing both the mycelium and AZ, the surfactin stayed on the same level as in the control (*p* ≤ 0.05).

In the case of iturin, it was found that compared to the 24th hour of cultivation, iturin levels increased more than 3-fold in the 48th and 72nd hour, with the highest production at the 48th hour for each cultivation variant (Figure 4b). A significant (*p* ≤ 0.05) influence of both the fungus and AZ on the synthesis of this lipopeptide was found. Compared to control, at the 48th hour of cultivation a 21.7% increase in iturin production was noted for the culture containing the bacteria and AZ. No negative effect of the mycelium presence on bacterial iturin production was observed. In fact, in the 48th hour of cultivation a 17 mg L^−1^ increase (from 52.5 to 69.5 L^−1^) in iturin levels in bacterial–fungal co-cultures was recorded. At every evaluated stage of cultivation, the highest iturin levels were noted in bacterial–fungal co-cultures with the addition of AZ (24.1, 78.1, and 68.8 mg L^−1^ at the 24th, 48th and 72nd hour, respectively). The highest overall iturin level was noted in the bacterial–fungal co-cultures with the addition of AZ in the 48th hour of cultivation (78.1 mg L^−1^).

One of the major explanations for the combined effect of biological and synthetic antifungal agents is that synthetic fungicides may alter the production of lipopeptide antibiotics such as surfactin, iturin, and fengycin. There have been reports of synergistic bioactive effects of different types of lipopeptide biosurfactants produced by *Bacillus* bacteria. In a study conducted by Wang and co-workers [44], the authors reported enhanced activity of surfactin and fengycin. They found that surfactin played an auxiliary role by improving the inhibitory effect of fengycin B against *Phytophthora infestans* mycelium growth. Similarly, Mihalache and co-workers [45] revealed an enhanced antifungal effect of mycosubtilin when it was used in combination with surfactin. Contrary to our results, Xu and co-workers [36] found that surfactin production was stimulated by the fungicide difenoconazole.

As was previously mentioned, iturin is a biosurfactant exhibiting strong antifungal activity. To the best of our knowledge, there have been no studies of AZ influence on iturin production by *Bacillus* bacteria, even though there have been several studies on improved bacilli action against phytopathogenic fungi in the presence of different synthetic fungicides, including strobilurins [31,32,33,34,35].

The data presented in Figure 4 might be pointing towards a potential shift in metabolic activities in *Bacillus* sp. Kol B3 when incubated together with *F. sambucinum* IM 6525. Specifically, we observed that while the presence of the fungus had a negative effect on the production of surfactin, and no effect on the production of iturin, the additional presence of AZ caused the bacteria to produce more surfactin and iturin.

Based on the obtained data, it could be suggested that the changed profile of biosurfactant production increases the destabilization of fungal membranes and facilitates penetration of the synthetic fungicide into the hyphae, accelerating the negative effect of AZ on mitochondrial function.

### 2.6. Fungicide Removal

All the tested cultivation variants showed similar AZ levels (*p* ≤ 0.05) compared to their respective controls (Figure 5), with one exception. A 20% decrease in AZ was noted in 72 h cultivation containing both microorganisms. In cultures containing *F. sambucinum* IM 6525 (with and without the addition of the bacteria), in the 48th and 72nd hours of the process, the AZ metabolite named azoxystrobin free acid (identified according to the MRM pair *m*/*z* 390/344) was noted. The same compound was also described by Gautam and Fomsgaard [46] as one of AZ metabolites in greenhouse grown lettuce.

According to the review by Feng and co-workers [47], among strobilurin-degrading microbes, bacteria play the most critical role. Even though the *Bacillus* sp. Kol B3 strain examined in this study did not exhibit promising AZ degradation properties, the other strains belonging to the genus *Bacillus* had been previously reported to be able to remove strobilurin fungicides [48].

To the best of our knowledge, there are no reports on *Fusarium* strains being able to degrade or transform azoxystrobin. Considering that the *Bacillus* sp. Kol B3 bacteria did not show significant ability to remove AZ (*p* ≤ 0.05), it can be suggested that this bacterial strain might be used as an effective biocontrol agent in soils polluted with AZ. Moreover, those two agents might act synergistically against fungal phytopathogens.

## 3. Materials and Methods

### 3.1. Microbial Strains

The *Bacillus* sp. Kol B3 strain, isolated from *Vinca minor* L. rhizosphere in Kolnica village (Greater Poland voivodeship) in Poland, was used in the study. The strain was confirmed to be able to produce surfactin, iturin, and fengycin by using MALDI-TOF/TOF (Appendix A).

*Fusarium sambucinum* IM 6525, obtained kindly from Lidia Sas-Paszt (The National Institute of Horticultural Research, Skierniewice, Poland), was isolated from infected raspberry roots.

The strains are stored in the strain collection of the Department of Industrial Microbiology and Biotechnology, University of Lodz, Poland.

### 3.2. Identification of Bacterial Strain Kol B3

Extraction of DNA from bacterial colonies was carried out by means of a commercial GeneMatrix Bacterial & Yeast Genomic DNA Purification Kit for isolating DNA from bacteria and yeast (EURx). DNA concentration in the samples was measured with a spectrophotometer at λ = 260 nm. For further analyses, the samples were diluted to a final concentration of 10 ng/μL.

Identification of bacterial strain Kol B3 was based on the analysis of the *tuf* gene sequence. The *tuf* gene was amplified using tufGPF/tufGPR primers [49]. Reaction mixtures (20 μL) consisted of 1× buffer for PCR, 0.2 mM of each nucleotide, 0.2 μM of each primer, 0.5 U of DreamTaqTM polymerase (ThermoScientific^®^, Waltham, MA, USA) and 20 ng of DNA. Amplification of the *tuf* gene was carried out in 35 cycles (95 °C × 30 s, 55 °C × 1 min, and 72 °C × 30 s). The identification of the bacterial strain was based on comparison of the obtained sequences with NCBI data using the BLAST tool (National Center for Biotechnology Information, Bethesda, ML, USA). Based on the analysis of the *tuf* gene sequence, the bacterial isolate Kol B3 was found to belong to the genus *Bacillus* (Appendix A).

### 3.3. Chemicals

The fungicide used in the preparation of cultivations (Amistar^®^ 250 SC; 250 mg L^−1^ azoxystrobin) was obtained from Target S.A. (Kartoszyno, Poland). Surfactin, iturin and AZ standards, chemicals used for surfactants and fungicide extraction by QuEChERS (quick, easy, cheap, effective, rugged, and safe) techniques, propidium iodide, sodium dodecyl sulfate (SDS), HEPES and PBS buffers, and Luria–Bertani (LB) medium were obtained from Sigma-Aldrich (Darmstadt, Germany). Chemicals used in the analysis performed by the techniques of liquid chromatography with tandem mass spectrometry (LC-MS/MS) were purchased from Avantor Performance Materials Poland S.A. (Gliwice, Poland). The FUN™-1 Cell Stain used to determine fungal viability was purchased from Invitrogen (Carlsbad, CA, USA).

### 3.4. Submerged Cultures Preparation

Liquid LB medium was inoculated with 1% of 24 h old *Bacillus* sp. Kol B3 and 10% of 24 h old *F. sambucinum* IM 6525 liquid precultures (prepared in LB medium). Then, Amistar^®^ 250 SC was added to obtain the final concentration 250 µg L^−1^ of AZ. Appropriate biotic controls (bacterial or fungal cultures with and without AZ addition) and abiotic controls (with no microorganism addition) were also prepared. The cultures in the final volume of 30 mL were incubated for 24, 48, and 72 h in 100 mL Erlenmeyer flasks. All cultivations and all abiotic controls were incubated on a rotary shaker (120 rpm) at 28 °C.

### 3.5. Antifungal Activity Analysis

The antifungal activity was assessed based on the fungal biomass and expressed as a percentage of the fungal control culture. The biomass was separated from the extracellular fluid by vacuum filtration using previously weighed filter papers No. 1 (Sartorius, Göttingen, Germany). The mycelium was then dried at 60 °C to reach a constant weight.

### 3.6. Effect of AZ on Bacillus *sp.* Kol B3 Growth

The influence of AZ on *Bacillus* sp. Kol B3 growth was determined for 48 h liquid cultivations, prepared as described above. First, serial dilutions were prepared, and agar plates were inoculated with 100 µL of the final dilution. The plates were incubated for 48 h at 28 °C. Then, CFUs were counted. Additionally, the bacterial biomass was separated from the supernatant by vacuum filtration using previously weighed filter papers (Sartorius, Germany). The bacteria were then dried at 60 °C to reach a constant weight.

### 3.7. Biosurfactant Extraction and Analysis

Surfactin and iturin were isolated by the QueChERS technique and analysed quantitively by LC-MS/MS, as described by Walaszczyk and co-workers [50]. Fengycin production was confirmed by MALDI-TOF/TOF techniques according to the method described by Paraszkiewicz and co-workers [51].

### 3.8. Fungicide Extraction and Analysis

The fungicide AZ and its metabolites were extracted using the modified QueChERS method described by Nykiel-Szymańska and co-workers [52]. Cultivations (20 mL) were transferred into 50-mL Falcon tubes. Next, glass beads and 10 mL of acetonitrile were added. The cultivations were then homogenized twice for 4 min at 25 m s^−1^ (Retsch, Ball Mill MM 400). To the obtained homogenate, four salts (2 g of MgSO_4_, 0.5 g of NaCl, 0.5 g of C_6_H_5_NaO_7_ × 2H_2_O, and 0.25 g of C_6_H_6_Na_2_O_7_ × 1.5H_2_O) were added and the extraction was carried out using the QuEChERS method. Then, the sample was centrifuged for 5 min at 2000× *g* and the top layer was collected for LC-MS/MS analysis. Measurement of AZ was performed using the ExionLC AC UHPLC system (Sciex, Framingham, MA, USA) and a 4500 Q-Trap mass spectrometer (Sciex, USA) with an ESI source. For reversed-phase chromatographic analysis, 5 µL of the lipid extract was injected onto a Kinetex C18 column (50 mm × 2.1 mm, particle size: 5 μm; Phenomenex, Torrance, CA, USA). The mobile phase consisted of 5 mM ammonium formate in water (A) and 5 mM ammonium formate in methanol (B). The solvent gradient was initiated at 20% B, increased to 90% B in 2 min and was maintained at 90% B for 2 min, before returning to the initial solvent composition after 2 min. The column temperature was maintained at 35 °C and the flow rate was 500 µL min^−1^. Nitrogen was applied as a curtain gas, a nebulising gas, and a turbo gas. The instrumental settings for tandem mass spectrometry were as follows: spray voltage 5500 V; curtain gas (CUR) 25; nebulizer gas (GS1) 50; turbo gas (GS2) 50; and ion source temperature 500 °C. Data analysis was performed with the Analyst tm v1.6.3 software (Sciex, USA).

Tandem mass spectrometry for the identification as well as quantification of AZ species was performed using standard and multiple reaction monitoring (MRM) pairs *m*/*z* 404/372, 404/344.

### 3.9. Fungal Membrane Permeability Assay

To assess fungal membranes permeabilization in the presence of the *Bacillus* sp. Kol B3 bacteria and the AZ fungicide, a method described previously by Litwin and co-workers [53] was used. Briefly, (1) 1 mL of liquid cultivation was added to an Eppendorf tube and centrifuged for 10 min at 12,000 rpm. (2) The supernatant was removed followed by the addition of 1 mL of PBS and 2 µL of propidium iodide (0.1 mg/mL concentration). (3) The sample was vortexed for 30 s and incubated in darkness for 5 min. (4) The sample was centrifuged again, the supernatant was removed, and 1 mL of PBS was added. The process was repeated twice. (5) The sample was placed on a 24-hole plate and fluorescence was measured at λex = 540 nm and λem = 630 nm. The fluorescence of the supernatant was also measured in order to control for the noise in the sample. (6) The fungal biomass from the sample was filtrated, dried, and weighted. The results were expressed as fluorescence intensity divided by a milligram of dried biomass.

### 3.10. Microscopic Analysis

The evaluation of the microscopic morphology of the fungi was carried out after 72 h of cultivation. Cultures were harvested and spread on a microscope slide. Microscopic images were captured using an LSM 5 (Zeiss, Oberkochen, Germany) confocal laser scanning microscope equipped with a Nomarski DIC and Plan-Neofluar 40× objective lenses.

### 3.11. Fungal Viability

From 72 h cultures, 1 mL was collected and centrifuged (8000× *g*). The supernatant was discarded and the mycelium was resuspended in 1 mL of 10 mM HEPES buffer containing 2% glucose and centrifuged again. The mycelia were then resuspended in 0.5 mL of HEPES buffer and 1 µL of FUN-1 dye was added. After 15 min incubation in the dark, microscopic preparations were made. Photographs of the fungal cells were taken using an Axiovert 200 M fluorescence microscope (Zeiss), excitation485/emission 505 and 560. The viability was calculated as a ratio between red and green fluorescence multiplied by 100. The average of all images obtained for the control was taken as 100%. Also, the results were presented as microphotographs.

### 3.12. Measurement of Intracellular Peroxynitrate Anion/Hydroxyl Radical Anion

The measurement of the overall intracellular levels of peroxynitrate anion/hydroxyl radical anion (HO•/ONOO^−^) was conducted using the 2′,7′–dichlorodihydrofluorescein diacetate (H_2_DCFDA) assay, as previously described by Nykiel-Szymańska and co-workers [54]. Briefly, 1 mL of 24 h long *F. sambucinum* IM 6525 cultivations were centrifuged at 10,000× *g* for 10 min. Then, the supernatant was discarded and the biomass was re-suspended in 1 mL of phosphate-buffer (PBS) containing 50 mM H_2_DCFDA (previously dissolved in dimethyl sulfoxide, DMSO). The samples were left to incubate for 15 min in the dark at room temperature. Next, the mycelium was washed twice with PBS and was then placed on a microscope slide. Imaging was performed using a confocal laser scanning microscope (LSM510 Meta, Zeiss) in conjunction with an Axiovert 200 M (Zeiss, Germany) inverted fluorescence microscope equipped with a Plan Apochromat objective (63/1.25 oil). Excitation was achieved using a 488 argon laser line, and the emission spectrum was recorded using a 530 nm bandpass filter. The results were expressed as a percentage of the green fluorescence area relative to the total hyphal area.

### 3.13. Statistical Analysis

All experiments were conducted at least in triplicate. The obtained results were presented as their mean values. An average standard deviation (SD) was calculated. One-way ANOVA analysis of variance was performed using Microsoft^®^ Excel^®^ for Microsoft 365 MSO, version 2401 (Microsoft Corporation, Washington, DC, USA). Differences at *p* ≤ 0.05 were considered significant.

## 4. Conclusions

*Fusarium* fungi pose a significant threat to global agriculture, causing devastating diseases in various crops. In this study, we investigated the combined antifungal effect of the commonly used fungicide azoxystrobin (AZ) in combination with biosurfactant producing *Bacillus* sp. Kol B3 bacteria against the phytopathogenic *Fusarium sambucinum* IM 6525 strain. The aim of the study was to evaluate how the *Bacillus* sp. Kol B3 bacteria and AZ work together against the fungus and gain deeper understanding of the mechanisms involved in antifungal activity. It has been suggested in the past that combining synthetic chemicals with biological control agents could reduce the rates of synthetic pesticide application [55,56].

The obtained results showed an increased (*p* ≤ 0.05) antifungal effect when using both fungicidal agents together. This effect was evident through fungal growth inhibition, changes in hyphae morphology, increased fungal membrane permeability, and modifications of intracellular ROS levels. Interestingly, while AZ did not significantly affect surfactin production, it had a mitigative effect on the decrease in surfactin levels in the presence of mycelium. In contrast, iturin production was not negatively affected by the presence of mycelium, and the addition of AZ resulted in a notable increase in the iturin levels at subsequent cultivation stages.

The findings highlight the complex interactions between synthetic fungicides, biosurfactant producing bacteria, and fungal phytopathogens. Notably, the protective effect of AZ on the surfactin levels in the presence of fungi as well as the positive influence of AZ on iturin production provide potential avenues for optimizing biosurfactant production in the presence of both synthetic fungicides and phytopathogenic fungal strains.

The presented data revealed the potential of biosurfactant producing bacteria to improve the antifungal effect of strobilurin fungicides. Considering all of the results, it is reasonable to speculate that *Bacillus* sp. Kol B3 and AZ might enhance each other’s antifungal activity against the studied fungus. However, proving that enhancement or synergism is responsible for this phenomenon requires further investigation.

## Figures and Tables

**Figure 1 ijms-25-04175-f001:**
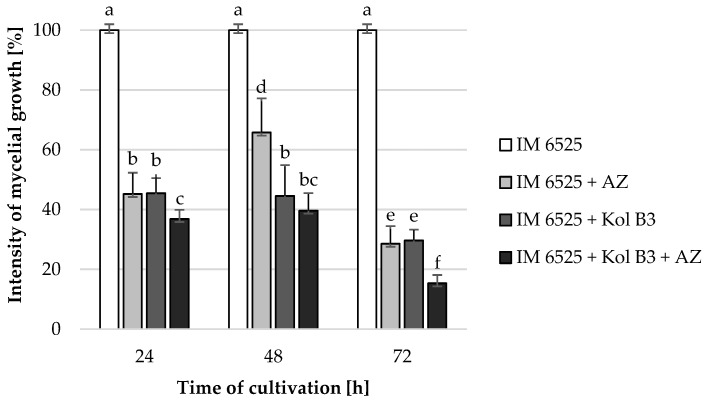
*F. sambucinum* IM 6525 growth intensity [%] in LB medium after 24, 48, and 72 h of cultivation. IM 6525—control fungal culture; IM 6525 + AZ—fungal culture amended with azoxystrobin; IM 6525 + Kol B3—bacterial-fungal co-culture; IM 6525 + Kol B3 + AZ—bacterial–fungal co-culture amended with azoxystrobin. The significance of the differences between the samples was determined according to ANOVA tests (*p* ≤ 0.05) and denoted by letters from “a” to “f”.

**Figure 2 ijms-25-04175-f002:**
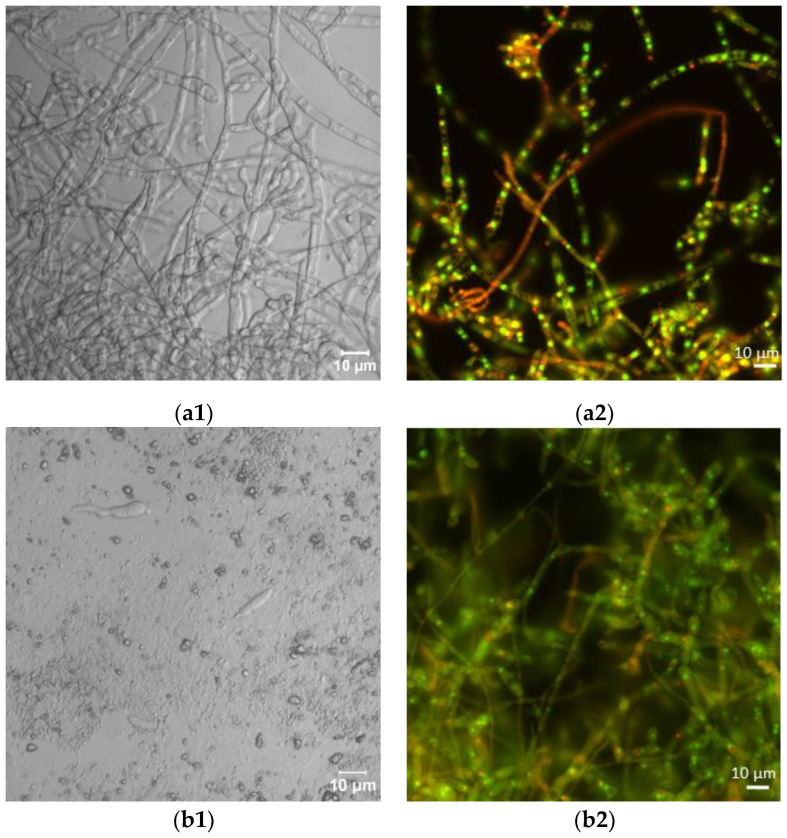
Antifungal activity after 72 h of cultivation. (**a1**,**a2**) *F. sambucinum* IM 6525 (control); (**b1**,**b2**) *F. sambucinum* IM 6525 + azoxystrobin; (**c1**,**c2**) *F. sambucinum* IM 6525 + *Bacillus* sp. Kol B3; and (**d1**,**d2**) *F. sambucinum* IM 6525 + *Bacillus* sp. Kol B3 + azoxystrobin. Pictures (**a1**–**d1**) were taken using a confocal laser scanning microscope. Pictures (**a2**–**d2**) were taken using a fluorescence microscope and the fungal mycelia were stained with a FUN-1 cell stain. Green stained mycelia are dead while alive cells generate red-fluorescent intravacuolar structures.

**Figure 3 ijms-25-04175-f003:**
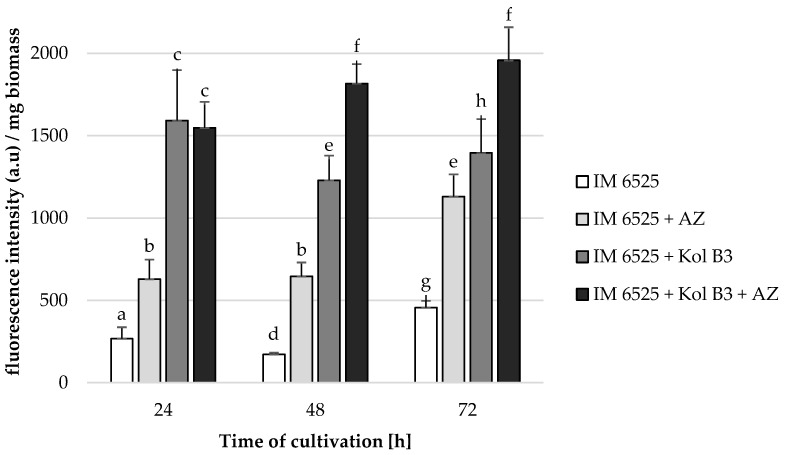
*F. sambucinum* IM 6525 membrane permeabilization [%] in LB medium after 24, 48 and 72 h of cultivation. IM 6525—control fungal culture; IM 6525 + AZ—fungal culture amended with azoxystrobin; IM 6525 + Kol B3—bacterial-fungal co-culture; IM 6525 + Kol B3 + AZ—bacterial–fungal co-culture amended with azoxystrobin. The significance of the differences between the samples was determined according to ANOVA tests (*p* ≤ 0.05) and denoted by letters from “a” to “h”.

**Figure 4 ijms-25-04175-f004:**
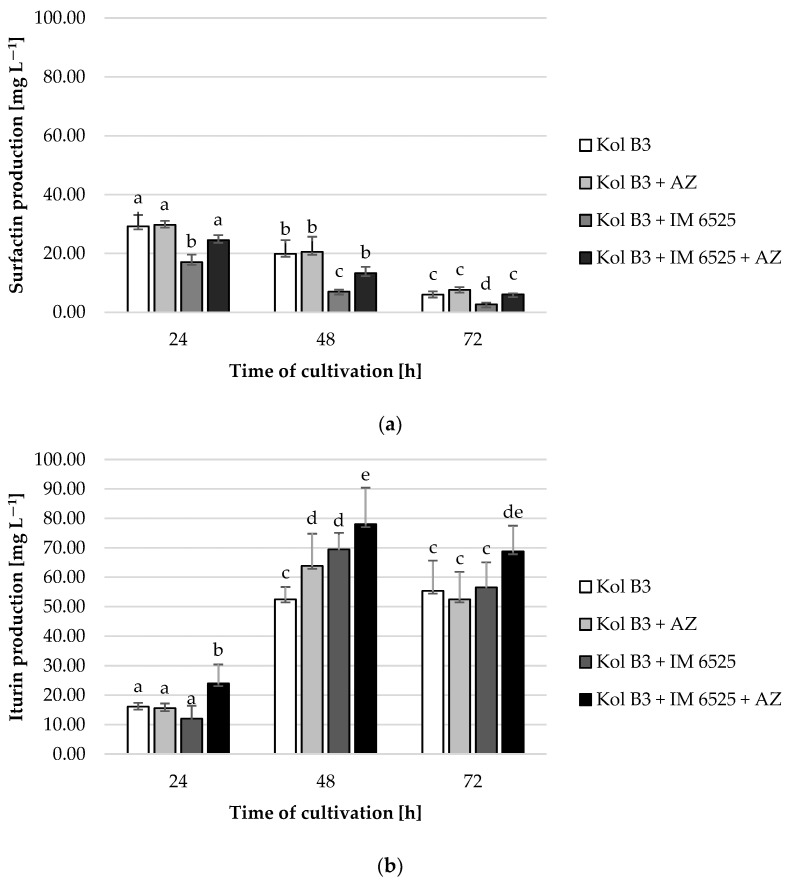
Surfactin (**a**) and iturin (**b**) production by *Bacillus* sp. Kol B3 in the presence of *F. sambucinum* IM 6525 and/or AZ. Kol B3—bacterial control culture; Kol B3 + AZ—bacterial culture amended with azoxystrobin; Kol B3 + IM 6525—bacterial–fungal co-culture; Kol B3 + IM 6525 + AZ—bacterial–fungal co-culture amended with azoxystrobin. The significance of the differences between the samples was determined according to ANOVA test (*p* ≤ 0.05) and denoted by letters from “a” to “e”.

**Figure 5 ijms-25-04175-f005:**
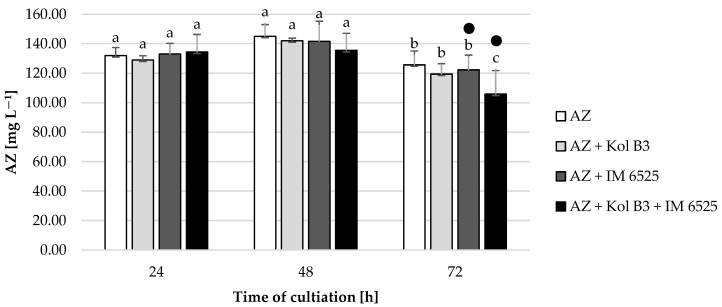
Fungicide AZ removal by *Bacillus* sp. Kol B3 and *F. sambucinum* IM 6525 fungi. AZ—azoxystrobin solution in LB medium, control; AZ + Kol B3—bacterial cultivation amended with azoxystrobin; AZ + IM 6525—fungal cultivation amended with azoxystrobin; AZ + Kol B3 + IM 6525—bacterial–fungal co-culture amended with azoxystrobin. The dot (●) symbol marks the cultivation variants containing AZ metabolite—azoxystrobin free acid. Letters from “a” to “c” indicate the values that differ significantly from each other at *p* ≤ 0.05.

**Table 1 ijms-25-04175-t001:** *F. sambucinum* IM 6525 viability after 72 h of cultivation in the presence of AZ and/or *Bacillus* sp. Kol B3 bacteria. Data are presented as the means ± SDs. The significance of the differences between the samples was determined according to ANOVA test (*p* ≤ 0.05). * indicates the values that differ significantly from the controls.

Cultivation Variant	*F. sambucinum* IM 6525 Viability [%]
IM 6525 (control)	93.14 ± 6.46
IM 6525 + AZ	5.03 ± 1.49 *
IM 6525 + Kol B3	0.18 ± 0.08 *
IM 6525 + AZ + Kol B3	0.00 ± 0.00 *

IM 6525—control fungal culture; IM 6525 + AZ—fungal culture amended with azoxystrobin; IM 6525 + Kol B3—bacterial–fungal co-culture; IM 6525 + Kol B3 + AZ—bacterial–fungal co-culture amended with azoxystrobin.

**Table 2 ijms-25-04175-t002:** Intracellular ROS generation (HO•/ONOO^−^) in *F. sambucinum* IM 6525 after 24 h exposure to AZ and/or *Bacillus* sp. Kol B3. The results are expressed as a percentage of the green fluorescence area compared to the total hyphal area. Data are presented as the means ± SDs. The significance of the differences between the samples was determined according to ANOVA test (*p* ≤ 0.05). * indicates the values that differ significantly from the control.

Cultivation Variant	HO•/ONOO^−^ [%]
IM 6525 (control)	0.02 ± 0.03
IM 6525 + AZ	3.99 ± 1.38 *
IM 6525 + Kol B3	0.77 ± 0.14 *
IM 6525 + AZ + Kol B3	0.04 ± 0.10

IM 6525—control fungal culture; IM 6525 + AZ—fungal culture amended with azoxystrobin; IM 6525 + Kol B3—bacterial–fungal co-culture; IM 6525 + Kol B3 + AZ—bacterial–fungal co-culture amended with azoxystrobin.

## Data Availability

The data presented in this study are available on request from the corresponding author.

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
