# Peer review of "The Combined Effects of Azoxystrobin and the Biosurfactant-Producing Bacillus sp. Kol B3 against the Phytopathogenic Fungus Fusarium sambucinum IM 6525"

_ijms, 2024, doi:10.3390/ijms25084175_

Round 1

Reviewer 1 Report

Comments and Suggestions for Authors

Corrections required

Title: change “biosurfactant producing Bacillus sp.” to “biosurfactant produced by Bacillus sp.”

 Specific comments

The paper is well written and contains interesting information.

However, it needs major corrections, mainly due to it is built on conceptual errors. See below.

The main drawback of this work is that the authors use the word ‘synergistic’ without having performed any assay to determine synergism.

The assessing of combinations is not an easy task. It requires a very careful design, determination of indices such as Fractional Inhibitory Combination Index (FICI), Combination Index (CI) or other, and a very precise interpretation of results by indicating the range of the indices for the interpretation of results.

The most common design for assessing synergy is the checkerboard method.

With this method, you can determine the FICI. Please read Vitale, R.G., Afeltra, J. and Dannaou, E., 2005. Antifungal combinations. In: Methods in Molecular Medicine, Vol. 118: Antifungal Agents: Methods and Protocols pages 143-152. Edited by: E.J. Ernst and P.D. Rogers © Humana Press Inc., Totowa, NJ, USA.

For interpretation of the results and FICI calculation, see Subheading 3.1., step 8 (page 147). Also read Odds, 2003. Synergy, antagonism, and what the chequerboard puts between them J. Antimicrob. Chemother. 52, 1

The following paper uses the checkerboard method with filamentous fungi.

Drogari-Apiranthitou et al., 2012. In vitro antifungal susceptibility of filamentous fungi causing rare infections: synergy testing of amphotericin B, posaconazole and anidulafungin in pairsJ Antimicrob Chemother. 67, 1937-1940.

See also Table 1 in Heyn et al., 2005. Effect of voriconazole combined with micafungin against Candida, Aspergillus, and Scedosporium spp. and Fusarium solani. Antimicrob. Agents Chemother. 49, 5157-5159.

There are other methods apart from the checkerboard. You can use the CompuSyn software and determine the CI. A CI <1, =1 and >1 indicates synergism, additive effect and antagonism, respectively. Read Eid et al. 2012. Synergism of three-drug combinations of sanguinarine and other plant secondary metabolites with digitonin and doxorubicin in multi-drug resistant cancer cells. Phytomedicine 19, 1288-1297. Read the section “Analysis of the combination effects” and Table 1.

In fact, the authors found “enhancements”. But this enhancement must be quantified too.

For example, the authors should have used the Dose Reduction Index (DRI) values. Also read Eid et al., Phytomedicine 2012 for DRI definition and interpretation of results.

Please note that in line 106, you say: “significantly increased (by half) antifungal activity was observed”. How much it increased? What is the DRI? Support the word “significant” with the “p” value.

Eliminate the word Synergistic from the title, line 212 and others.

Other remarks

The section Methods is missing in the Abstract

Lines16, 89: in its first mentioning, the binomial name must be written in full: Change “F. sambucinum” to “Fusarium sambucinum”. Note that the Abstract and the main text, each works on its own.

In lines 65-67, you say: “There are three families of cLP biosurfactants that can be produced by Bacillus bacteria: surfactins, iturins and fengycins [16,17]. Surfactin shows mainly antiviral and antibacterial properties, while iturin and fengycin exhibit mainly antifungal activity [14]. Please note that at first you named the compounds in plural (surfactins, iturins, fegycins) , but then, in singular (surfactin, iturin, fegycin). Please clarify. Note that surfactin is a cyclic lipopeptide, commonly used as an antibiotic for its capacity as a surfactant. So, it is “A COMPOUND”. Regarding Iturin, do you refer to ‘iturin A’? Fengycn is a cyclic lipopeptide.

Line 75: change “are not without their” to “have several”

Line 83: change “activity” to “activities”

Lines 461-463: You say: “The obtained results showed a significantly ENHANCED antifungal effect when using both fungicidal agents together. This effect was evident through STRONG fungal growth INHIBITION, changes in HYPHAE MORPHOLOGY, INCREASED fungal membranes PERMEABILITY and MODIFICATIONS of intracellular ROS levels. The obtained results clearly showed that combining Bacillus sp. Kol B3 bacteria and AZ could SIGNIFICANTLY IMPROVE ANTIFUNGAL EFFICACY compared to the individual application of each agent.

As a summary, the authors must deepen the concept of synergism by reading many papers dealing with this subject.

Once the issue is understood, the authors must change the manuscript accordingly

Meanwhile the paper cannot be accepted for publication in IJMS

Comments on the Quality of English Language

The English is good

Author Response

Dear Reviewer,

To begin with, we would like to thank you for the effort in reviewing our manuscript. We will answer your comments point by point.

  1. “Title: change “biosurfactant producing Bacillus” to “biosurfactant produced by Bacillus sp.””
  • After careful consideration of all the comments, we agree that the title should be changed. However, we do not agree with this specific recommendation. In the title we would like to emphasize that we used the whole Bacillus cultivations, not just the isolated biosurfactants. Considering all of your suggestions, we believe that the new title should be changed to “Fusarium sambucinum growth inhibition in the presence of biosurfactant producing Bacillus and synthetic fungicide azoxystrobin”.

  1. “The main drawback of this work is that the authors use the word ‘synergistic’ without having performed any assay to determine synergism.

The assessing of combinations is not an easy task. It requires a very careful design, determination of indices such as Fractional Inhibitory Combination Index (FICI), Combination Index (CI) or other, and a very precise interpretation of results by indicating the range of the indices for the interpretation of results.

The most common design for assessing synergy is the checkerboard method.

With this method, you can determine the FICI.

Please read Vitale, R.G., Afeltra, J. and Dannaou, E., 2005. Antifungal combinations. In: Methods in Molecular Medicine, Vol. 118: Antifungal Agents: Methods and Protocols pages 143-152. Edited by: E.J. Ernst and P.D. Rogers © Humana Press Inc., Totowa, NJ, USA.

For interpretation of the results and FICI calculation, see Subheading 3.1., step 8 (page 147).

Also read Odds, 2003. Synergy, antagonism, and what the chequerboard puts between them J. Antimicrob. Chemother. 52, 1

The following paper uses the checkerboard method with filamentous fungi.

Drogari-Apiranthitou et al., 2012. In vitro antifungal susceptibility of filamentous fungi causing rare infections: synergy testing of amphotericin B, posaconazole and anidulafungin in pairsJ Antimicrob Chemother. 67, 1937-1940.

See also Table 1 in Heyn et al., 2005. Effect of voriconazole combined with micafungin against Candida, Aspergillus, and Scedosporium spp. and Fusarium solani. Antimicrob. Agents Chemother. 49, 5157-5159.

There are other methods apart from the checkerboard. You can use the CompuSyn software and determine the CI. A CI <1, =1 and >1 indicates synergism, additive effect and antagonism, respectively. Read Eid et al. 2012. Synergism of three-drug combinations of sanguinarine and other plant secondary metabolites with digitonin and doxorubicin in multi-drug resistant cancer cells. Phytomedicine 19, 1288-1297. Read the section “Analysis of the combination effects” and Table 1.”

  • We would like to thank you for your very thorough and informative review. We absolutely agree that the word “synergism” was used incorrectly considering the lack of methodology specific to that phenomenon. In our study, we aimed to use alive Bacillus cells, not the isolated biosurfactants alone. We agree that in order to use the word “synergistic” we would need to use one of the methods you mentioned, however our intention was to get deeper insight into the interaction between the bacteria and the fungus in the presence of a synthetic fungicide.

  1. “In fact, the authors found “enhancements”. But this enhancement must be quantified too.

For example, the authors should have used the Dose Reduction Index (DRI) values.

Also read Eid et al., Phytomedicine 2012 for DRI definition and interpretation of results.

Please note that in line 106, you say: “significantly increased (by half) antifungal activity was observed”. How much it increased? What is the DRI? Support the word “significant” with the “p” value.”

  • We agree that in the case of our study, Dose Reduction Index should have been applied. Thank you very much for that suggestion. We may run additional experiments in order to incorporate it into our manuscript. In our estimation, this addition would take an additional month to execute. In fact, it was not our intention to assess quantitively the dose of biosurfactants and synthetic fungicide used in combination, but describing fungal growth and assessing the mechanisms involved especially in the Bacillus antifungal activity. Instead of additional experiments, we might underline the original idea behind our study in the text. We propose substituting the words such as “synergism” or “enhancement” with better suited words and phrases describing the observed phenomenon (e.g. combined, parallel action, concurrent action).
  • Of course we improved the references to the statistical analysis.

  1. “Eliminate the word Synergistic from the title, line 212 and others.”
  • The word synergistic in the context of describing our results was eliminated from the manuscript.

Other remarks

  1. The section Methods is missing in the Abstract
  • The mistake has been corrected.

  1. Lines 16, 89: in its first mentioning, the binomial name must be written in full: Change “ sambucinum” to “Fusarium sambucinum”. Note that the Abstract and the main text, each works on its own.
  • The mistake has been corrected.

  1. In lines 65-67, you say: “There are three families of cLP biosurfactants that can be produced by Bacillus bacteria: surfactins, iturins and fengycins [16,17]. Surfactin shows mainly antiviral and antibacterial properties, while iturin and fengycin exhibit mainly antifungal activity [14]. Please note that at first you named the compounds in plural (surfactins, iturins, fegycins) , but then, in singular (surfactin, iturin, fegycin). Please clarify. Note that surfactin is a cyclic lipopeptide, commonly used as an antibiotic for its capacity as a surfactant. So, it is “A COMPOUND”. Regarding Iturin, do you refer to ‘iturin A’? Fengycn is a cyclic lipopeptide.
  • The naming mistake has been corrected.
  • In detail, each of Bacillus biosurfactant family (surfactin, iturin or fengycin) is a mixture of cyclic lipopeptides built by variants of a peptides and a β-hydroxy fatty acid with chain length of different number of carbon atoms. Therefore, the compounds might be produced as different homologues (differing in the length of the fatty acid chain) as well as different isoforms (differing in the amino acid composition of the peptidic sequence). Of course, iturin A is one of the isoforms from the iturin family. In the Introduction section, we briefly describe the biosurfactants produced by Bacillus

  1. Line 75: change “are not without their” to “have several”
  • The suggested change was made.

  1. Line 83: change “activity” to “activities”
  • The suggested change was made.

  1. Lines 461-463: You say: “The obtained results showed a significantly ENHANCED antifungal effect when using both fungicidal agents together. This effect was evident through STRONG fungal growth INHIBITION, changes in HYPHAE MORPHOLOGY, INCREASED fungal membranes PERMEABILITY and MODIFICATIONS of intracellular ROS levels. The obtained results clearly showed that combining Bacillus sp. Kol B3 bacteria and AZ could SIGNIFICANTLY IMPROVE ANTIFUNGAL EFFICACY compared to the individual application of each agent”.
  • We assume that this comment refers to too strongly worded conclusions. Therefore, as we mentioned before, we propose rephrasing the manuscript in order to better suit the achieved results. We especially focused on the cited paragraph.

We attached the improved manuscript. All of the changes can be viewed in review mode.

Kind regards,

Katarzyna Paraszkiewicz and co-workers

Reviewer 2 Report

Comments and Suggestions for Authors

The work is devoted to experimental testing of the idea that biofungicides can and should be used in combination with synthetic conventional fungicides. The idea itself is not new; manufacturers of microbiological fertilizers and biopesticides test them for compatibility with various synthetic pesticides. And farmers often substitute synthetic pesticides with biological products [10.3390/agriculture13081590 ]. Specifically, in the present work, the authors indicate the purpose of their study to evaluate the effect of a bacterial producer of biosurfactants combined with azoxystrobin on the viability of Fusarium. In the introduction, I missed a paragraph describing well-known examples of combining biological products and synthetic pesticides: what is the practice in the world, maybe statistical data emphasizing the importance of such research, etc.

Despite the fact that the results of the work are predictable (lipopeptide metabolites of bacilli in combination with AZ will enhance each other’s effects), I did not find any serious methodological flaws, and some of the results are quite interesting (the effect of AZ on the production of iturins and surfactins).

2.1. This section can be moved to the Suppl. files. This section contains only technical information. I would like to clarify why the authors carried out identification using one gene (tuf) if the result (unambiguous identification) could not be achieved?

Instead of this section, add a description of the results of the analysis of metabolites of Bacillus sp. Kol B3 (the section 2.6 instead of 2.1).

Figure 1-4. Somehow indicate in the figure the p-values when using Anova (approximately the same as in Figure 5).

L234-235. Existing researches show that reactive oxygen species, which damage many intracellular macromolecules, determine the “indirect mode of action.”

The section 3.9. Fungal membrane permeability assay.

This assay is missing two controls (positive and negative). I recommend that you still indicate which device was used to assess fluorescence.

L 396. Is the flow exactly 500 ml per minute? Check it

L 397. Clarify that “The instrumental settings” were for MS…

What does the abbreviation "QueChERS" technique mean?

Author Response

Dear Reviewer,

To begin with, we would like to thank you for the effort in reviewing our manuscript. We will answer your comments point by point.

  1. The work is devoted to experimental testing of the idea that biofungicides can and should be used in combination with synthetic conventional fungicides. The idea itself is not new; manufacturers of microbiological fertilizers and biopesticides test them for compatibility with various synthetic pesticides. And farmers often substitute synthetic pesticides with biological products [3390/agriculture13081590 ]. Specifically, in the present work, the authors indicate the purpose of their study to evaluate the effect of a bacterial producer of biosurfactants combined with azoxystrobin on the viability of Fusarium. In the introduction, I missed a paragraph describing well-known examples of combining biological products and synthetic pesticides: what is the practice in the world, maybe statistical data emphasizing the importance of such research, etc.
  • An appropriate paragraph was added to the Introduction section.

  1. Despite the fact that the results of the work are predictable (lipopeptide metabolites of bacilli in combination with AZ will enhance each other’s effects), I did not find any serious methodological flaws, and some of the results are quite interesting (the effect of AZ on the production of iturins and surfactins).
  • Thank you for your kind opinion. To our best knowledge, there are no studies available on azoxystrobin’s influence on surfactin and iturin produced by bacilli.

  1. 1. This section can be moved to the Suppl. files. This section contains only technical information. I would like to clarify why the authors carried out identification using one gene (tuf) if the result (unambiguous identification) could not be achieved?
  • We agree with the recommendation of moving section 2.1 to the supplementary files.
  • Considering the degree of similarity between different Bacillus subtilis species complex, the differentiating between them is a very difficult task. In order to achieve an unambiguous identification, not only the whole genome sequencing, but also average amino acid identity, average nucleotide identity and digital DNA-DNA hybridization and other tests should be performed. The results obtained by us confirmed only that the bacterial strain used in the study belongs to subtilis species complex, specifically to B. subtilis or B. amyloliquefaciens group. For the purposes of this study, we decided that more detailed identification was not needed.

References:

  • Fan B., Blom J., Klenk H. P., Borriss R. 2017. Bacillus amyloliquefaciens, Bacillus velezensis, and Bacillus siamensis Form an “Operational Group amyloliquefaciens” within the B. subtilis Species Complex. Frontiers in Microbiology, 8:22.
  • Auch A. F., von Jan M., Klenk H. P., Göker M. 2010. Digital DNA-DNA hybridization for microbial species delineation by means of genome-to-genome sequence comparison. Standards in Genomic Science, 2:117-34.
  • Ngalimat M. S., Yahaya R. S. R., Baharudin M. M. A., Yaminudin S. M., Karim M., Ahmad S. A., Sabri S. 2021. A Review on the Biotechnological Applications of the Operational Group Bacillus amyloliquefaciens. Microorganisms, 9:614.

  1. Instead of this section, add a description of the results of the analysis of metabolites of BacillusKol B3 (the section 2.6 instead of 2.1).
  • We considered implementing your suggestion. Initially we agreed, however, after deeper consideration, we believe that moving subsection “Biosurfactant production” to the beginning of the “Results and discussion” section will not be an improvement. In our manuscript, we wanted to first describe the antifungal activity of Bacillus Kol B3 and azoxystrobin and then move on to discussing possible mechanisms involved in the antifungal activity. Considering that the biosurfactant production by Bacillus sp. Kol B3 is one of those mechanisms, we think its description should stay after the description of antifungal activity.

  1. Figure 1-4. Somehow indicate in the figure the p-values when using Anova (approximately the same as in Figure 5).
  • The suggested change was made.

  1. L234-235. Existing researches show that reactive oxygen species, which damage many intracellular macromolecules, determine the “indirect mode of action.”
  • Thank you very much for the suggestion. Of course, the increased ROS levels might indeed damage many intracellular macromolecules, including compounds building cell membranes, resulting in changes in the fluidity and physical state of the structure. We modified the mentioned paragraph in order to emphasize this fact.

  1. The section 3.9. Fungal membrane permeability assay. This assay is missing two controls (positive and negative). I recommend that you still indicate which device was used to assess fluorescence.
  • In our manuscript, the negative control are untreated fungal cells described as “IM 6525 – control fungal culture” on the Figure 3. We might run additional experiments in order to add the positive control to our manuscript. We estimate that it would require an additional month before submitting the final version of the paper. However, we believe that this addition will not significantly improve our paper, since the differences between our cultivation variants are evident and statistically significant.

  1. L 396. Is the flow exactly 500 ml per minute? Check it
  • Of course an error have been made. It was suppose to be 500 µL, not mL. The error has been corrected.

  1. L Clarify that “The instrumental settings” were for MS…
  • The suggestion has been implemented.

  1. What does the abbreviation "QueChERS" technique mean?
  • The abbreviation QueChERS means “quick, easy, cheap, effective, rugged, and safe”. The explanation of the abbreviation was added in subsection 3.3 “Chemicals”, when it first appears in the text.

We attached the modified manuscript. All of the changes can be viewed in review mode.

Kind regards,

Katarzyna Paraszkiewicz and co-workers

Reviewer 3 Report

Comments and Suggestions for Authors

The article by Walaszczyk et al., describes the effects of fungicide azoxystrobin and Bacillus producing biosurfactants on FusariumFusarium sambucinus is a phytopathogen causing crop diseases resulting in compromised global food security and economy. The article is an interesting read. Here are my comments:

1) Fig 1: The authors should mention, which growth rates are signifiantly different from each other within the time points in the figure. For example, is there any statistical difference between the mycelial growth between the control and under AZ at 24h?

2)Please mention the statistical tests performed in tables 2 and 3 in the table legend for better clarity.

3)Figs 3 and 4: The authors should mention, which results are significantly different within the time points in the figure. 

4)Fig 5: What are the circular dots on the bars? Are those supposed to be asterix (*)? Please clarify

Author Response

Dear Reviewer,

To begin with, we would like to thank you for the effort in reviewing our manuscript. We will answer your comments point by point.

  1. Fig 1: The authors should mention, which growth rates are significantly different from each other within the time points in the figure. For example, is there any statistical difference between the mycelial growth between the control and under AZ at 24h?
  • The suggested change was made.

  1. Please mention the statistical tests performed in tables 2 and 3 in the table legend for better clarity.
  • The appropriate addition was made in legends for the Tables mentioned.

  1. Figs 3 and 4: The authors should mention, which results are significantly different within the time points in the figure. 
  • The suggested change was made.

  1. Fig 5: What are the circular dots on the bars? Are those supposed to be asterix (*)? Please clarify
  • As is mentioned in the description of Figure 5, the circles/dots symbols mark the cultivation variants containing AZ metabolite. As for significant differences, appropriate denotations in the Figure were added.

We attached the modified manuscript. All of the changes can be viewed in review mode.

Kind regards,

Katarzyna Paraszkiewicz and co-workers

Reviewer 4 Report

Comments and Suggestions for Authors

In the paper entitled "Synergistic antifungal effect of azoxystrobin and biosurfactant producing Bacillus sp. on phytopathogenic Fusarium sambucinum strain", the authors present the synergistic antifungal effects of a synthetic fungicide (AZ) and the biosurfactant producing Bacillus sp. Kol B3.

I must say that it was a real pleasure reviewing this paper. It is well designed, well written and very clear to various readers. Also, the idea of combining two types (natural and synthetic) fungicides is a great one and the methods applied are accurate and correctly described.

In my personal opinion, this article can be accepted for publication, after some very minor changes:

- I was wondering if paragraphs on page 2 (second one, 2.2. Antifungal activity) - lines 116-147 could not be reduced to one paragraph only...I believe it is too long for a scientific article, and more suitable for a review article;

-page 4, line 184: replace "nonspecific" with "non-specific";

- page 7 - line 295: the is a missing dot (.) at the end of the paragraph;

- page 7 - line 299: F. sambucinum should be in Italic face;

-page 8 - lines 312 and 314: Bacillus should be in Italic face;

- page 11 - lines 457-458: the authors' statement "The aim of the study was to evaluate how the presence of the fungicide may influence Bacillus antifungal activity".....shouldn't it be the other way around? that the addiction of Bacillus sp. Kol B3 enhances the activity of AZ, as the authors very nicely explain on page 7 lines 292-295? it is true that the aim of the study can be one, but the result can be different....but the conclusion should always come based on the results obtained.

Author Response

Dear Reviewer,

To begin with, we would like to thank you for the effort in reviewing our manuscript. We are very glad you liked our manuscript. We will answer your suggestions point by point.

  1. I was wondering if paragraphs on page 2 (second one, 2.2. Antifungal activity) - lines 116-147 could not be reduced to one paragraph only...I believe it is too long for a scientific article, and more suitable for a review article;
  • We agree with your suggestion. The mentioned text was shortened and reduced to one paragraph only.

  1. page 4, line 184: replace "nonspecific" with "non-specific";
  • The suggested change was made.

  1. page 7 - line 295: the is a missing dot (.) at the end of the paragraph;
  • The error has been corrected.

  1. page 7 - line 299: F. sambucinum should be inItalic face;
  • The error has been corrected.

  1. page 8 - lines 312 and 314: Bacillus should be inItalic face;
  • The error has been corrected.

  1. page 11 - lines 457-458: the authors' statement "The aim of the study was to evaluate how the presence of the fungicide may influence Bacillusantifungal activity".....shouldn't it be the other way around? that the addiction of Bacillus Kol B3 enhances the activity of AZ, as the authors very nicely explain on page 7 lines 292-295? it is true that the aim of the study can be one, but the result can be different....but the conclusion should always come based on the results obtained.
  • We agree with the suggestion that the aims should be phrased similarly throughout the manuscript in order to avoid confusion. In fact, our intention was not to study the influence of AZ on Bacillus bacteria or the other way around, but to assess how the two agents will work together against the fungus. We made appropriate changes in the manuscript.

We attached the modified manuscript. All of the changes can be viewed in review mode.

Kind regards,

Katarzyna Paraszkiewicz and co-workers

Round 2

Reviewer 1 Report

Comments and Suggestions for Authors

The authors improved the manuscript and have now avoided the use of the word 'synergism'.

Regarding the title:

The authors did not use in the title the words “combined” or “combination”. Thus, the title does not clearly reflect the content of this work, which is based on the anti-Fusarium sambucinum activity of the combinations of AZ with the biosurfactant produced by Bacillus sp. Kol B3.

Please read the first lines of the Abstract: “The study aimed to evaluate how the combined presence of synthetic fungicide azoxystrobin (AZ) and biosurfactant producing Bacillus sp. Kol B3 influences the growth of the phytopathogenic fungus Fusarium sambucinum IM 6525.

A clear title should be: “Effect of the combined effects of azoxystrobin (AZ) and the biosurfactant producing Bacillus sp. Kol B3, against the phytopathogenic fungus Fusarium sambucinum IM 6525”

Regarding the work

The idea of combining biopesticides and synthetic pesticides as practical approach to pest management is good. However, the effect of the combinations must be demonstrated in a proper way.

Aim

Authors say in lines 108, 109: “The aim of the study was to compare the impact of AZ, biosurfactant producing Bacillus sp. Kol B3 as well as these two agents combined on Fusarium sambucinum IM 6525 109 mycelium.”

Conclusions

In the Conclusions, the authors state in lines 625, 626: “Considering all of the results, it is reasonable to SPECULATE that Bacillus sp. Kol B3 and AZ MIGHT enhance each other’s antifungal activity against the studied fungus. However, proving that enhancement or synergism is responsible for this phenomenon requires FURTHER INVESTIGATION.”

My opinion is that the authors would have tested both partners with a proper design, they could have determined if both partners showed synergism, and in what ratio. Within the ratio, in which concentrations. Instead, they performed a study that, as they state in lines 625, 626, “requires further investigation”

I recommend the reading of the following papers.

-         Cuenca Estrella, 2004. Combinations of antifungal agents in therapy-what value are they? J. Antimicrob. Chemother. 54, 854–869 .

-         Vitale, R.G., Afeltra, J. and Dannaou, E., 2005. Antifungal combinations. In: Methods in Molecular Medicine, Vol. 118: Antifungal Agents: Methods and Protocols pages 143-152. Edited by: E.J. Ernst and P.D. Rogers © Humana Press Inc., Totowa, NJ, USA.

-         Eid et al., 2012. Synergism of three-drug combinations of sanguinarine and other plant secondary metabolites with digitonin and doxorubicin in multi-drug resistant cancer cells Phytomedicine 19, 1288–1297

The authors performed other studies but they should have performed them on the most effective combination, but this ratio was not determined.

The Editor must pass judgement if the revised manuscript is of interest for the audience. This Reviewer cannot recommend this paper, although revised, for publication in IJMS. The studies of combinations require proper designs and proper interpretations of the results.

Comments on the Quality of English Language

The English is good

Author Response

Dear Reviewer,

Thank you very much for all of your comments.

We changed the title according to your suggestion.

As for the other comments, we would like to underline the fact that in our research we wanted to check whether the combined effect will be present at all and investigate some of the possible mechanisms of antifungal activity involved, especially the biosurfactants production. We do plan on continuing the research further in the direction suggested (determining the dose and synergism but also further exploring the interactions at hand between the bacteria, fungi and azoxystrobin).

We wrote to the Editor to let them decide on the fate of the manuscript.

Kind regards,

Katarzyna Paraszkiewicz and co-workers

Round 3

Reviewer 1 Report

Comments and Suggestions for Authors

The authors have now meet the requirements of this Editor. The paper can be accepted.